# Conceptual Design and Performance Optimization of a Tip Device for a Regional Turboprop Aircraft

**Ilias Lappas** [1,*,†] **and Akira Ikenaga** [2,†]

1    Department of Mechanical and Aeronautical Engineering, School of Engineering, University of South Wales, Treforest Campus, Treforest CF37 1DL, UK
2    Future Projects Department, Airbus UK, Filton, Bristol BS34 7PA, UK; akira.ikenaga@airbus.com
*    Correspondence: ilias.lappas@southwales.ac.uk; Tel.: +44-01443-482565
†    These authors contributed equally to this work.

**Abstract:** An increasing number of aircraft is equipped with wing tip devices, which either are installed by the aircraft manufacturer at the production line or are retrofitted after the delivery of the aircraft to its operator. The installation of wing tip devices has not been a popular choice for regional turboprop aircraft, and the novelty of the current study is to investigate the feasibility of retrofitting the British Aerospace (BAe) Jetstream 31 with an appropriate wing tip device (or winglet) to increase its cruise range performance, taking also into account the aerodynamic and structural impact of the implementation. An aircraft model has been developed, and the simulated optimal winglet design achieved a 2.38% increase of the maximum range by reducing the total drag by 1.19% at a mass penalty of 3.25%, as compared with the baseline aircraft configuration. Other designs were found to be more effective in reducing the total drag, but the structural reinforcement required for their implementation outweighed the achieved performance improvements. Since successful winglet retrofit programs for typical short to medium-range narrow-body aircraft report even more than 3% of block fuel improvements, undertaking the project of installing an optimal winglet design to the BAe Jetstream 31 should also consider a direct operating cost (DOC) assessment on top of the aerodynamic and structural aspects of the retrofit.

**Keywords:** winglets; lift-induced drag; wing tip device; non-planar lifting surface; performance optimization; aircraft performance; regional aircraft; turboprop aircraft

## 1. Introduction

The aerospace industry relies on the continuous improvement of existing technologies and on the innovative development of new concepts to improve the return of investment of an aircraft design. Many of these improvements from the airframe manufacturer's perspective are focused on the aerodynamic efficiency and the structural design. Aligned to the public's increasing awareness of the environmental footprint of the airline industry and to the uncertainty in airline revenue stemming from the fluctuating oil prices, there has been a push toward developing eco-friendly designs. Governmental bodies and the aerospace industry have set targets and road maps to improve current airframes and develop new technologies in power plant design and air traffic management to reduce the environmental impact of air transportation systems. According to a study commissioned by the Department of Transport [1], winglets are a valid NOx and $CO_2$ abatement technology that can result in a reduction of 2% in fuel burn with relatively small capital investment and modification cost. A similar conclusion has been reached from the study of Farriers and Eyers [2], in which winglets are considered as an evolutionary airframe technology to provide an overall aerodynamic advantage to previous and current generation airframes.

### 1.1. Induced Drag Reduction Techniques

The induced drag coefficient $C_{D,i}$ is defined as:

$$C_{D,i} = \frac{C_L^2}{\pi e AR} \tag{1}$$

in which $C_L$ is the coefficient of lift, $e$ is the Oswald efficiency factor, and $AR$ is the aspect ratio. In theory, minimum induced drag can be achieved by increasing the geometrical span to infinity (infinite $AR$) and by optimizing the spanload to match the elliptical distribution ($e = 1$). Unfortunately, the induced drag reduction does not necessarily translate directly to better aircraft performance in terms of range increase or fuel saving when compared against the baseline aircraft configuration, as is the case for a retrofit study. A span extension reduces induced drag and increases the wetted area, therefore increasing profile drag; hence, there exists a crossover point at which increasing span is no longer beneficial.

From the structural perspective, the ideal spanload may not result in an efficient structural design due to the forces and moments experienced by the structure. As a certain proportion of the wing structural weight is related to resisting bending stress, an increase of bending moment is considered to increase wing weight [3]. Therefore, the aerodynamic efficiency resulting from induced drag reduction should be analyzed in the context of both the parasitic drag and the structural analysis considerations for the configuration in question, and their overall effect on the aircraft performance needs to be evaluated. The aerodynamic and structural objectives work against each other, and a meta objective becomes necessary to evaluate the effects at the aircraft level rather than the component level. After Prandtl's [4] formulation of the lifting line theory and the associated induced drag, there have been practical studies [5,6] on minimizing induced drag by studying the effects on spanload (spanwise distribution of lift) and accounting for the bending moment experienced by the wing as a measure of the structural impact.

### 1.2. Non-Planar Devices

Even before the Wright brothers' first powered flight in 1903, English aerodynamicist Lanchester patented what is now known as end plates, and described the function of the truncated wings fitted with vertical capping wings as "*to minimize the loss of energy due to air circulation around the wind* (sic: wing) *extremities*" [7]. More recently, Whitcomb [8] studied the use of smaller wing extensions attached to the wing tips on transonic aircraft and called them "winglets". Further studies have been carried out [9–15], in which various methods were used to assess the aerodynamics and the assumed structural impact of adding non-planar span extensions, although with no clear consensus regarding the advantage of a non-planar extension over planar extensions. Assessing the aerodynamics of the different wing options, several methods were used, from fully numerical [9] to purely experimental [8,10]. As highlighted by Asai [11], the computer models that have been used to analyze drag were lacking viscous drag measurements. The most widely used method in assessing the structural impact measures is the wing-root bending moment (WRBM), but that does not accurately represent the loads transmitted on the structure, such as torsion or inertia moments. Elham and Van Tooren [14] addressed this issue by using medium fidelity tools, but still allowing for minimized computational cost when analyzing a wide design space.

### 1.3. Winglet Studies on Various Aircraft Platforms

Typical missions flown by large transonic aircraft include a lower stratospheric cruise segment lasting an average of 90% of the block time, in which induced drag accounts for about 25% of total drag [16]. There is a limited number of studies on wing-tip implementation or research for relatively smaller turboprop aircraft, and especially regional commuters. A summary of those studies is provided below.

The United States Air Force (USAF) has commissioned the National Research Council to evaluate its aircraft inventory and identify those aircraft that may be good candidates for winglet modifications [17]. The C-130 turboprop tactical airlifter has been evaluated as well, but concerns were highlighted on the suitability of the aircraft for a winglet retrofit, as the wing is already very efficient due to its relatively high aspect ratio ($AR$ = 10.1) and non-swept wings with low loaded wing tips. These two factors compounded by the short operating missions and the low flight altitudes minimize the benefits of winglets, as their optimum flight condition is in higher altitudes for higher wing-tip loadings and longer cruise segments in lower density air.

Lehmkuehler and Wong [18] have designed winglets for the Fairchild Merlin III turboprop eight-seat commuter aircraft, and their Computational Fluid Dynamics (CFD) analysis has shown a 5% gain in the lift-to-drag ratio (L/D) over the reference wing. The study has focused on the aerodynamic design using CFD, without performing a detailed study on the structural impact of the winglet. Nicolosi et al. [19] have designed a twin-engine 11-seat commuter aircraft incorporating the aerodynamic optimization of the wing using winglets with the purpose of increasing take-off, climb, approach, and landing performance. Della Vecchia [20] has developed methods for aerodynamic design and optimization aimed at developing a new regional turboprop aircraft using the ATR 72 aircraft as a baseline. His configuration has used optimized winglets that have shown an improved L/D, including shorter take-off runs, a faster rate of climb, a higher operating ceiling, and lower fuel consumption for a given mission.

The literature review yielded no previous study on an aircraft similar in configuration to the BAe Jetstream 31. Lehmkuehler and Wong [18] and Della Vecchia [20] have developed winglets for a smaller and larger aircraft respectively in terms of maximum take-off weight (MTOW), with limited consideration on the structural effects of a tip device. The structural impact was evaluated by comparing WRBMs, and this is an approach that has been considered as not capturing all the potential structural considerations [11,13,14]. The practical effects of the improvements achieved by the use of winglets have been summarized by Conley [21] and Dees and Stowell [22] as:

- Decrease in fuel burn (or increase in range).
- Increase in flight ceiling.
- Reduced take-off runs.
- Increased time between engine maintenance.

### 1.4. General Framework for the Use of Winglets in Regional Turboprop Aircraft

An overview of regional turboprop aircraft offering a capacity of 19 passengers is presented in the Table 1.

**Table 1.** Overview of regional turboprop aircraft (19 passengers capacity). (Data sourced from the respective aircraft manufacturer brochures, not an exhaustive list of all operating types). MTOW: maximum take-off weight.

| Type | Year of First Flight | Maximum Operating Altitude (ft) | Range (Km) | MTOW (Kg) | Aspect Ratio | Winglets Use |
|---|---|---|---|---|---|---|
| DHC-6 Twin Otter (Series 400) | 2010 | 25,000 | 1480 | 5670 | 10.0 | No |
| Harbin Y-12 | 1982 | 23,000 | 1340 | 5300 | 8.7 | No |
| Beechcraft 1900D | 1982 | 25,000 | 707 | 7764 | 10.8 | Yes |
| Dornier Do-228 | 1981 | 25,000 | 396 | 6575 | 9.0 | No |
| BAe Jetstream 31 | 1980 | 25,000 | 1260 | 6950 | 10.0 | No |
| Let L-410 Turbolet | 1969 | 27,500 | 510 | 6600 | 11.5 | No |
| Fairchild Swearingen Metroliner | 1969 | 25,000 | 2131 | 6577 | 10.5 | No |
| Embraer EMB 110 Bandeirante | 1968 | 21,500 | 1964 | 5900 | 8.1 | No |

Table 1 is a representative market overview of various aircraft that are still in service and offer a capacity of 19 passengers. It is observed that the majority of the aircraft have at least two common characteristics: Their first flight was more than 35 years ago, and they are not equipped with winglets.

The absence of winglets in this aircraft category further reinforces the USAF's conceptual context that has highlighted concerns regarding the suitability of the C-130 fleet for a winglet retrofit [17], taking into account the high AR values, the short operating missions, and the low cruise flight altitudes. Those factors form the rational of the framework of Table 1.

Regarding the operational aspects of the framework, the DHC-6 offers a very similar range to the Jetstream 31 with a 1-ton lower MTOW, and has a very similar range performance in comparison to the Chinese Harbin Y-12. The Jetstream 31 would have outperformed the Y-12 in terms of range with a 6% increase of its current range. Furthermore, the only aircraft that is currently in production is the DHC-6 Twin Otter, and as such, a potential winglet retrofit of the Jetstream 31 might provide an incentive to maintain it in service for longer, if it proved to be worth undertaking. It is considered that the incentive to keep the Jetstream 31 in service is significant, since there are only a few options for an aircraft of similar characteristics. The widespread availability of the Jetstream 41, a stretched and re-engined variant, can also provide common spare parts to the Jetstream 31 to sustain the existing fleet, thus justifying further investment into a Jetstream 31 airframe with performance improvements. For all the above reasons, the application of a winglet retrofit for the Jetstream 31 has been undertaken, and it is presented in the sections that follow.

The BAe Jetstream 31 is a regional turboprop derived from the earlier Handley Page HP.137 Jetstream. Designed for regional routes, it is a small twin-engine turboprop with pressurized fuselage carrying from 12 to 19 passengers. The Jetstream 31 was designed for a niche market aimed at airlines wishing to offer regional commuter service at higher speeds between small regional airports. A wing-tip retrofit option entails maintaining the original wing shape and structure as close as possible, while achieving a performance improvement with the installation of an efficient tip device. There are differing views on the feasibility of wing-tip treatments for short domestic flights, as the aerodynamic improvements may not have a net fuel usage improvement (or range trade-off) due to the multidisciplinary nature of the modification. The design and implementation of winglets has been to this day a challenging multidisciplinary effort [23].

## 2. Methodology

### 2.1. Geometry Modeling

The aerodynamic effects of winglets and their advantages have been a contentious topic with no clear winner in terms of drag reduction compared to a planar extension, with most studies evaluating solely the induced drag and ignoring viscous effects. Therefore, the design scope of the present study includes planar extensions as well as non-planar devices based on Whitcomb's winglet. Figure 1 and Table 2 illustrate the design variables that have been chosen for evaluation, together with their respective values.

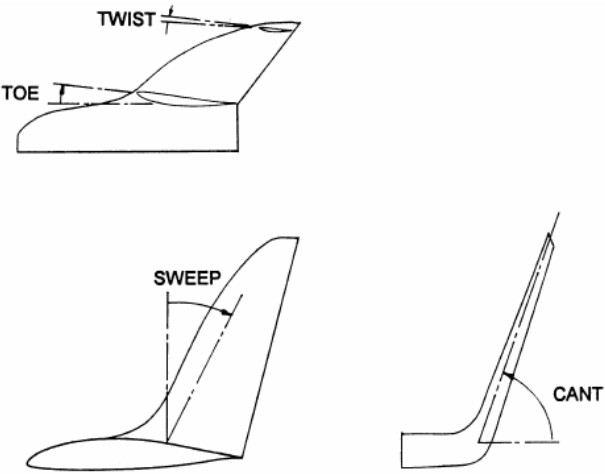

**Figure 1.** Basic winglet parameters terminology (from Maughmer [24]).

**Table 2.** Design variables and evaluated values (1296 design points).

| Variable (Unit) | Values | Justification/Constraints |
|---|---|---|
| Winglet position | 0.8, 0.98 | As a ratio of wing-tip chord. It was to either start from the leading edge (0.98) or recessed rearwards to minimize the wetted area, but still in front of the front spar position to facilitate integration to the front spar in subsequent detailed design. |
| Winglet taper ratio | 0.25, 0.5 | Combination of Lehmkuehler's [18] and Della Vecchia [20] optimal taper ratios. |
| Cant angle (absolute, from XY plane) (deg) | 7, 43.5, 80 | Planar 7 degrees, same as the main wing dihedral. An 80-degree maximum angle, with 43.5 being the middle value of the range. |
| Winglet span (or winglet height, as a ratio of wing semispan) | 10%, 15%, 20% | As a ratio of the wing semispan with the upper boundary constrained by the Jetstream 31 certification airport reference code (B-II), stipulating a maximum 24 m of span. The upper boundary was scaled down to a more reasonable maximum 20% of semispan. |
| Quarter chord sweep (deg) | 14, 26, 38, 50 | Although mostly suitable as a parameter for aircraft operating in the transonic regime, it is kept to investigate the aerodynamics and the torsional effects on the wing structure. |
| Toe-out (deg) | −4, −2.5, −1 | Toe-out angle dictated by the local lift coefficient requirement. |
| Twist (deg) | −4, −2.5, −1 | Twist controls the winglet spanload [20]. |

A Design of Experiment was set featuring a full factorial on the design parameters, yielding 1296 design points. The analysis of the design points was automated using Python (v2.7.13, released on 17 December 2016. Python Software Foundation, Wilmington, DE, USA). As the study compares a reference and project aircraft configuration, an initial reference model was created, verified, and validated. Then, the reference model was modified with specific design values to yield the project aircraft for each design point. OpenVSP (v3.10, released on 8 January 2017. NASA, Washington, DC, USA) was used for the geometry modeling of the aircraft. The benefits from using OpenVSP were the existing integration of the VSPAero aerodynamic solver based on the vortex lattice method (VLM), which was used for the purposes of this study, as well as the capability of meshing outputs for various finite element analysis packages. The geometry information has been obtained from aircraft manufacturer data [25] and previous simulation work performed by Cooke [26]. Aircraft components not modeled for this study are: nacelles, propellers, aft fuselage vertical strakes, landing gear doors, belly fairing, fuselage-wing root fairing, and vertical tail plane fillet. Actuator discs were modeled instead of propellers.

2.1.1. Reference Model

Figures 2–4 show the reference aircraft model overlaid on the aircraft maintenance manual [25] views.

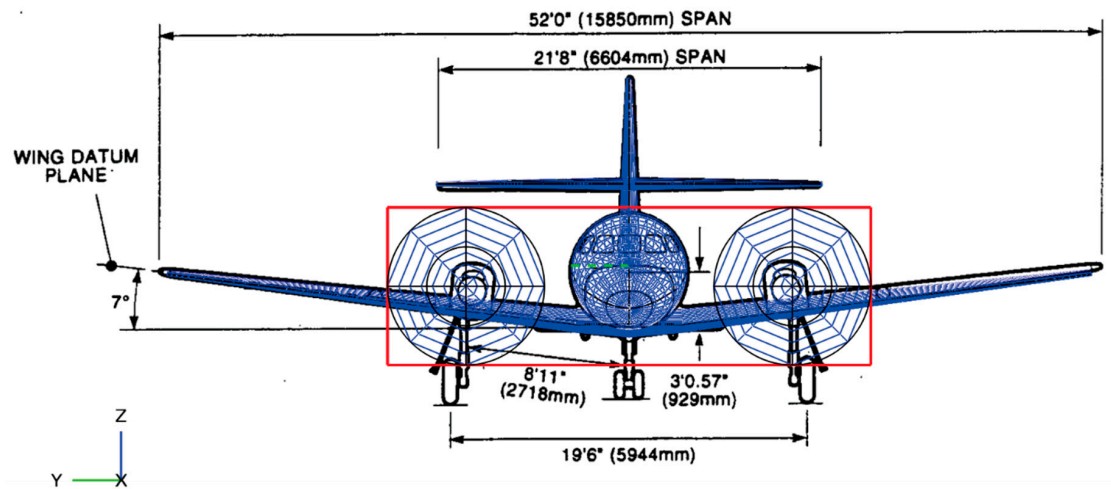

**Figure 2.** Reference model front view (overlaid on the aircraft maintenance manual [25] view).

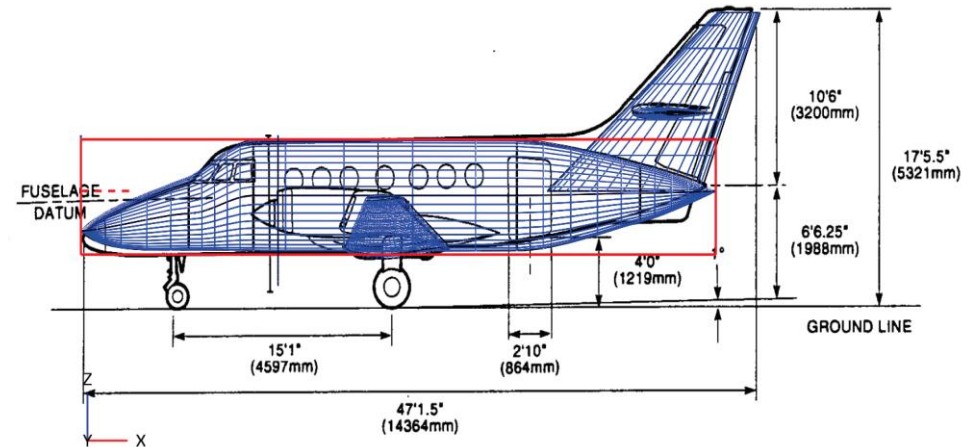

**Figure 3.** Reference model side view (overlaid on the aircraft maintenance manual [25] view).

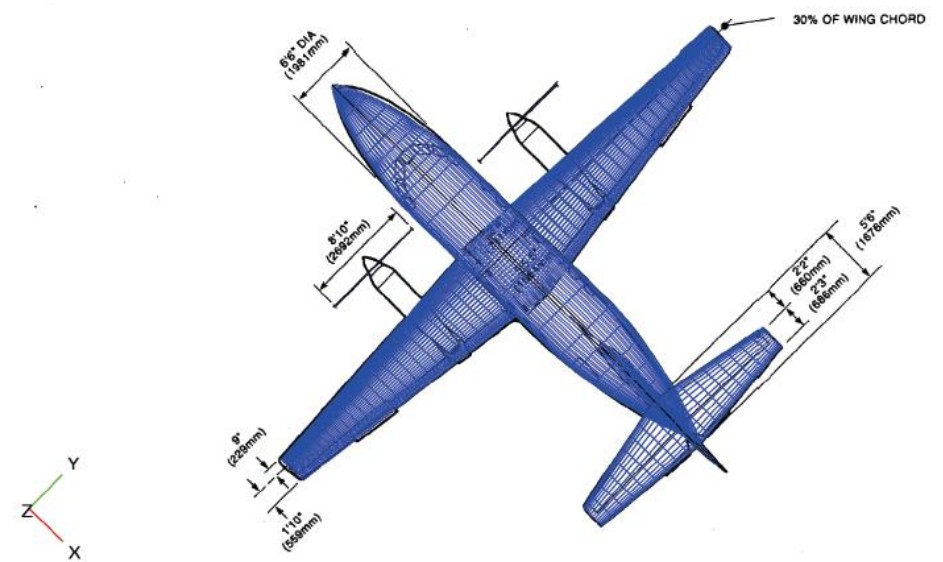

**Figure 4.** Reference model top view (overlaid on the aircraft maintenance manual [25] view).

### 2.1.2. Wing

The main wing was modeled as two panels, each one extending from the wing tip to the wing root position. The local axis of the wing is defined as the leading edge at the centerline. The six-digit NACA aerofoil coordinates were generated in OpenVSP, and the specific 'a' value defining the laminar proportion of the chord required for these types of aerofoil was set to 0. The dihedral was assumed to be specified along the mean camber line of the wing (planar surface created from each wing section data). The geometrical reference data for the wing is provided in Table 3.

**Table 3.** Geometrical reference data for the wing (geometry information from [25,26]).

| Parameter (Unit) | Value |
|---|---|
| Projected span (m) | 15.85 |
| Area (m2) | 25.902 |
| Centerline chord (m) | 2.375 |
| Root chord (m) | 2.16 |
| Tip chord (m) | 0.79 |
| Sweep at 30% of chord (deg) | 0 |
| Aspect ratio | 10.0 |
| Reference chord: Mean aerodynamic chord (MAC) (m) | 1.87 |
| Root setting angle (deg, applied at the 25% of the chord) | 3 |

Two extra panels were added to the wing: one transition panel after the wing tip, and a winglet panel. The purpose of using the transition panel was to allow the application of a toe-out angle to the wing-tip base. Applying the toe-out angle at the wing tip without this additional transition panel would have effectively changed the washout angle from the root, affecting the aerodynamic characteristics. Furthermore, on a retrofit study, the inner wing geometrical shape is assumed to be kept constant, or it would become a wing redesign project. For the lofting of the transition panel and winglet, an aerofoil designed specifically for winglets with low Reynolds numbers was chosen, the PSU-94-097 winglet aerofoil designed by Maughmer et al. [27], which has been also analyzed by Della Vecchia [20]. The transition panel was specified with a constant span of 0.1 m (independent of the winglet span), and the dihedral value of this transition panel was 67% of the winglet cant angle.

### 2.1.3. Fuselage

The fuselage was modeled with the 'FUSELAGE' component with a circular section for the center section between the cockpit and the exit door, and elliptical sections were used to model the cockpit, nose cone, and tail cone. The geometrical reference data for the fuselage is shown in Table 4.

**Table 4.** Geometrical reference data for the fuselage (geometry information from [25,26]).

| Parameter (Unit) | Value |
|---|---|
| Circular maximum diameter (m) | 1.981 |
| Length (m) | 13.347 |

### 2.1.4. Vertical and Horizontal Tailplanes

Both vertical and horizontal tailplanes "WING" sections were modeled with the same method as the main wing, with the vertical tailplane differing in that it was rotated 90 degrees to align with the Z axis. The aerofoils for the tail planes were generated in the OpenVSP NACA four-digit aerofoil generator tool. The geometrical reference data for the horizontal and vertical tailplanes are presented in Tables 5 and 6, respectively.

**Table 5.** Horizontal tail plane geometrical reference data (geometry information from [25,26]).

| Parameter (Unit) | Value |
| --- | --- |
| Projected span (m) | 6.60 |
| Area (m$^2$) | 7.80 |
| Centerline chord (m) | 1.676 |
| Tip chord (m) | 0.6855 |
| Sweep at 25% of chord (deg) | 7.10 |
| Aspect Ratio | 5.60 |
| Mean aerodynamic chord (MAC) (m) | 1.181 |
| Root aerofoil | NACA 0012 |
| Tip aerofoil | NACA 0010 |
| Dihedral (deg) | 0 |

**Table 6.** Vertical tail plane geometrical reference data (geometry information from [25,26]).

| Parameter (Unit) | Value |
| --- | --- |
| Projected span (m) | 3.32 |
| Area (m$^2$) | 6.65 |
| Centerline chord (m) | 3.20 |
| Tip chord (m) | 0.88 |
| Sweep at 25% of chord (deg) | 7.10 |
| Aspect Ratio | 5.60 |
| Mean aerodynamic chord (MAC) (m) | 2.04 |
| Root aerofoil | NACA 0012 |
| Tip aerofoil | NACA 0010 |

## 2.2. Aerodynamic Analysis

VSPAero, which is an integrated aerodynamic tool of OpenVSP, has been used. The 3D potential flow tool offers both VLM and panel implementations, with VLM simplifying the geometry to the mean camber lines, and the panel method representing the surface of the aircraft with vortex sheets. The VLM has been chosen, having considered the computational cost and the number of designs to be analyzed. The outcomes of the method were validated using published flight test data (the validation is analyzed in Section 2.3, below).

The cruise conditions of the aircraft model are shown in Table 7.

**Table 7.** Cruise reference conditions (geometry information from [25,26]).

| Parameter (Symbol) (Unit) | Value |
| --- | --- |
| Altitude (ft) | 25,000 |
| Ground speed ($V_\infty$) (km/h) | 425 |
| Mach number (M) | 0.38 |
| Density ($\varrho$) (International Standard Atmosphere (ISA), kg/m$^3$) | 0.5495 |
| Reynolds number (Re) | $7.98 \times 10^6$ (MAC) |
| Reference chord (m) | 1.87 (MAC) |
| Reference area ($S_{ref}$) (m) | 24.952 |
| Reference span (m) | 15.85 |

VSPAero does not predict viscous effects on lift and drag. XFoil (v6.99, released on 23 Dec 2013. Massachusetts Institute of Technology, Boston, MA, USA) was used to predict the maximum viscous lift coefficient $C_L$ at the operating conditions to constrain wing sectional lift coefficient $C_l$ to 1.49. At the cruise condition:

$$C_{Lcruise} = \frac{W_{cruise}}{\frac{1}{2}\rho V_\infty^2 S_{ref}} \tag{2}$$

The weight of the aircraft at cruise, $W_{cruise}$, can be approximated as [28]:

$$W_{cruise} = \sqrt{(MTOW) \; x \; (MZFW)} \tag{3}$$

From Equations (2) and (3), after substituting the approved values [29] for the MTOW and the maximum zero-fuel weight (MZFW), it is calculated that:

$$C_{Lcruise} = 0.66 \tag{4}$$

The actuator disks modeled in lieu of propellers modify the streamlines downstream of the disk to simulate prop wash and the contraction of streamlines. The model in VSPAero is based on Conway's [30] actuator disk theory, and it has been verified and validated for NASA projects such as Stoll's [31] investigation of the distributed propulsion blown wing. The strength of the actuator disk is determined by the thrust coefficient, the power coefficient, and the RPM, for which the default values in VSPAero of $C_T = 0.4$, $C_{Power} = 0.6$ and 2000 RPM, respectively, have been used.

The Certification Specification-25 (Large Aeroplanes) [32] specifies the certification requirements for which the sizing loads of the wing are derived from flight manoeuver and gust conditions. The critical sizing condition assumed for this study is a 2.5 g pull-up manoeuver at cruise condition, and VSPAero was run for both 1-g and 2.5-g conditions.

*2.3. Validation of Reference Aerodynamic Model*

Published flight test data for the Jetstream 31 of the National Flying Laboratory Centre (NFLC) [33] has been used to validate the VSPAero model. The actual flight test data have been best represented by the following equations for the lift and drag coefficients:

$$C_L = 0.3305 + 0.1052\alpha_b \tag{5}$$

$$C_D = 0.0376 + 0.0607C_L^2 \tag{6}$$

The applicability of Equations (5) and (6) is restricted to the linear portion of aerodynamics, since the NFLC aircraft is not allowed to stall; hence, the polars of the VSPAero model were only estimated for up to 10 degrees, while a design flight condition was chosen that mostly matches the theoretical lift coefficient value of 0.66 calculated above. The lift curve slope was found to be in good agreement with the flight test data, and a very good trend on the total drag model has been observed as well for low angles of attack, slightly diverging for values higher than 7 and up to 10 degrees. Table 8 shows the comparison between the predictions of the model and the published flight test data.

**Table 8.** Comparison with the flight test data [33]. L/D: lift-to-drag ratio.

| Parameter | Lawson et al. [32] | VSPAero Prediction for the Developed Model | % Difference |
|---|---|---|---|
| Lift curve slope | 0.1052 | 0.1101 | 4.66 |
| Lift coefficient for zero angle of attack | 0.3305 | 0.3323 | −2.48 |
| Lift coefficient at 3-degree angle of attack | 0.6461 | 0.655 | −1.4 |
| Zero lift drag coefficient | 0.0422 | 0.0433 | 0.26 |
| Drag coefficient at 3-degree angle of attack | 0.0629 | 0.0641 | 1.78 |
| L/D at zero angle of attack | 7.472 | 6.985 | −6.52 |
| L/D at 3-degree angle of attack | 10.27 | 10.23 | −0.4 |
| Maximum L/D | 10.5 | 10.78 | 2.66 |

*2.4. Structural Sizing*

Over the years, the development of novel and robust wing mass estimation methods has received significant attention [34]. Reliable and accessible wing mass prediction methods enable the preliminary assessment of the expected benefits of novel technologies, which can enhance the L/D of the aircraft wing. The student version of the Elham Modified Wing Weight Estimation Technique (EMWET) [35] has been used, which uses an analytical approach to size the wingbox primary structure, together with structural and material parameters, applied loads, and geometrical data of the wing planform and the related aerofoils. The technique achieves high levels of accuracy (average error on the total wing weight is consistently lower than 2%) and design sensitivity with low computational cost. By using the student version of EMWET, the accuracy of the results might slightly deviate from the mentioned above-average error [36]. Approved values from the aircraft type certificate [29] and its technical drawings [25] have been used as inputs to model the structure.

2.4.1. Reference Model Structural Validation

Published weight data for the Garret-TP331-10 engine [37] has been used. Structurally, the wingbox was simplified to two main spars (front and rear, the middle spar has been omitted). The fuel tank was modeled to occupy the complete wingbox volume enclosed between the front and rear spars. The rib pitch was approximated to 0.5 m per rib bay, with the stringers across the upper and lower covers to have an efficiency of 0.96 for 'Z'-type stringers [38]. The material defined for all panels is aluminum alloy 7075-T6. A kink was identified in the front spar after the power plant, and the structure was defined in four sections: center, root, kink, and tip. The calculated structural weight for the reference wing is 889.1 kg, which is 12.23% of the MTOW. By applying Elham's [38] wing weight estimation formula, which is based on statistical data and uses a power equation:

$$W_{wing \; (Elham \; eq.)} = 68.22 \times 10^{-4} MTOW^{1.25} \tag{7}$$

a wing weight $W_{wing}$ of 928.54 kg was estimated, which is 12.63% of the MTOW. This 0.4% difference can be attributed to the different aerodynamic methods used, as well as to the fact that the power equation does not contain any aircraft of comparable MTOW to the Jetstream 31. The smallest aircraft used in the validation of the power equation was the Fokker F50, which is an aircraft that is nearly three times heavier than the Jetstream 31. Furthermore, the student version of EMWET used for the calculations for both reference and project wing models relies on a different, simplified regression analysis when compared to the full version. Statistical data from Kundu [39] approximates the weight of the wing as 10–12% of MTOW for a mid-sized twin-engine turboprop aircraft, which is an estimate that is in agreement with the calculated value for the developed model.

2.4.2. Project Model Structural Design

For the definition of the winglet structure, an additional planform station was added at the winglet wing tip with a different airfoil (PSU-WL), and the spars were assumed to extend from the wing tip to winglet wing tip with identical front and rear spar locations as the wing tip (0.2 c, 0.74 c). Therefore, the winglet was treated as an extension of the original planform and an integral part of the wing. The wing tip-transition-winglet panel was simplified to only wing-tip and winglet wing-tip sections, since the winglet position was backwards at most until the front spar location (0.8 c). Figure 5 shows the top-view geometry of the new planform, following the addition of the winglet, in which Z-height distribution is the vertical height distribution.

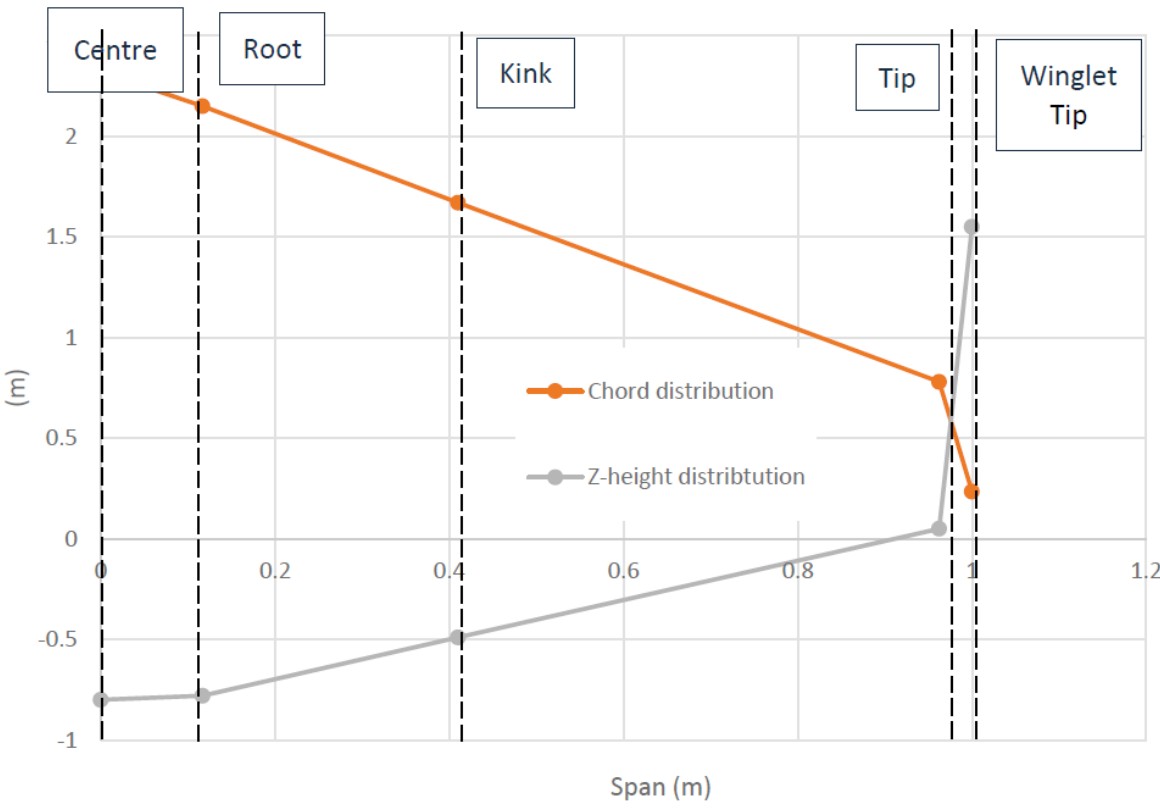

**Figure 5.** Reference model top view.

## 2.5. Aircraft Performance Assessment

The methodology followed by Mariens [36] and introduced by Roskam [40] is used to calculate the required fuel for the mission of the aircraft. This methodology uses the Breguet range equation, together with statistical factors that estimate the fuel weight of the typical segments of the aircraft flight mission. The statistical factors are shown in Table 9.

**Table 9.** Fuel fraction for each segment of a typical flight mission, as suggested by Roskam [40].

| Fuel Weight Fraction (Mffi) | Turboprop Aircraft |
| :---: | :---: |
| Start and warm up | 0.990 |
| Taxi | 0.995 |
| Take off | 0.995 |
| Climb | 0.985 |
| Cruise | Calculated |
| Descent | 0.985 |
| Landing, taxi, and shutdown | 0.995 |

Each fuel weight fraction $M_{ffi}$ indicates the ratio of the total aircraft weight at the end of the flight segment to the total aircraft weight at the beginning of the segment. Thus, the total fuel weight fraction defines the consumed fuel as a ratio of the total aircraft weight at the end of the flight mission to the total aircraft weight at the beginning. The total fuel weight fraction is also equal to the product of all the fuel weight fractions; thus, the following equation applies:

$$\mathrm{M}_{ff} = \prod_{i=1}^{n} M_{ffi} = 1 - \frac{W_{fuel}}{W_{take\ off}} \tag{8}$$

From Raymer [41], the propulsive efficiency of the model was assumed as $n_p$ = 0.8, while the specific fuel consumption is equal to $c_p$ = 9.344 × 10$^{-5}$ kg/[(Watt) × (sec)] [37]. Table 10 provides a synopsis of the weight data used for the performance calculations for the Jetstream 31.

**Table 10.** Synopsis of the weight data for the Jetstream 31 (cross-checked from [29,42]. MZFW: Maximum Zero-Fuel Weight. OEW: Operating Empty Weight.)

| Weights | Value (Kg) |
|---|---|
| Reference wing | 449.52 |
| MTOW | 7350 |
| MZFW | 6350 |
| Maximum payload | 1935 |
| Maximum fuel | 1491 |
| OEW | 4415 |

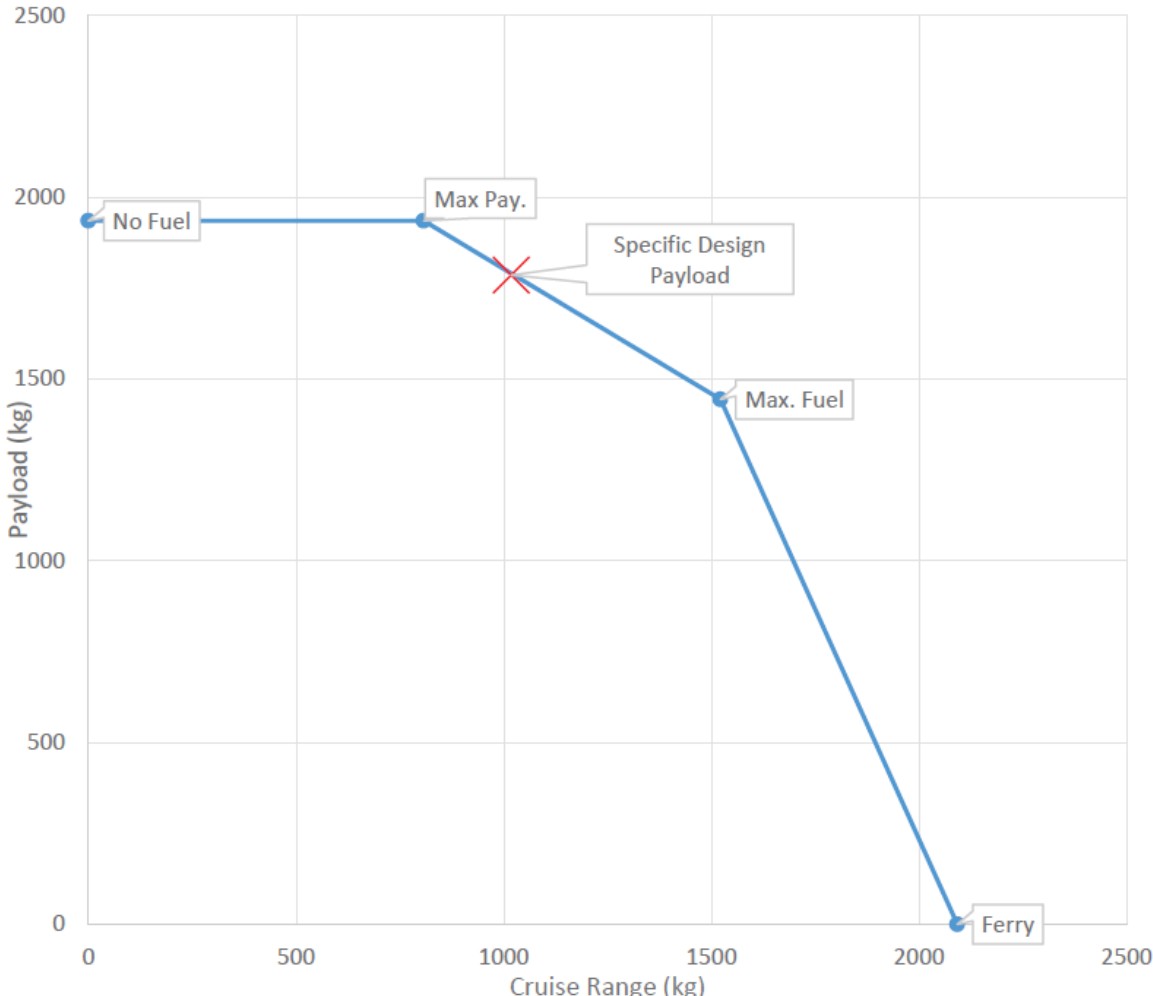

**Figure 6.** Payload-range diagram.

The payload-range diagram of the developed model is shown in Figure 6 above. The specific payload-range point is defined as the cruise range for a specified payload. Assuming 19 passengers at 94 kg per passenger (including luggage), the specified payload is 1786 kg with a calculated cruise range of 1018.15 km for the reference model aircraft. Then, the specific design point was assessed for each winglet permutation, resulting in L/D$_{project}$ and M$_{ffcruise(project)}$, the latter term being dependent on the OEW. The variation of the fuel weight affects the range of the aircraft; therefore, the structural impact on range can be assessed as well.

*2.6. Python Integration Framework*

Python has been used to create an automated integration framework. The Python environment was used to write OpenVSP script files to apply the design variables written also by Python from a separate file. The data generation workflow was split into the following major Python functions, while the flowchart of its implementation process is illustrated in Figure 7:

- OpenVSP and VSPAero Runner: Create a pool of design points for the desired range of variables. Then, this is fed to OpenVSP for each design point in an AngelScript++ input file for OpenVSP containing case-specific design parameters. Then, the geometry generated by OpenVSP was read in VSPAero and executed to calculate the aerodynamic data for 1 and 2.5-g conditions.
- EMWET input file parsing: As EMWET requires the geometry, spanload, and the quarter chord pitching moment, the VSPAero output files were parsed and written onto the EMWET initialization and load files. Each individual EMWET case was appended onto a Matlab × m script that could then be run from Matlab. To process the results, two functions were developed in Python to ultimately output a comma-separated values file in ASCII for data visualization and allow oprimization work in the future.
- Parsing of VSPAero aerodynamic coefficients and associated wing weight and joining each case input to its output.
- Writing case input and output in a *.csv file.

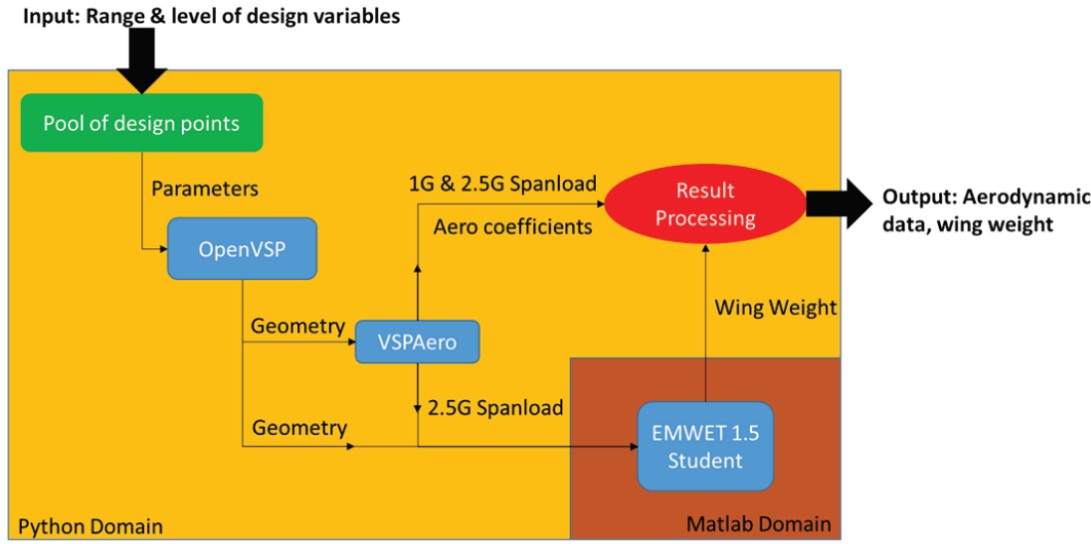

**Figure 7.** Implementation flowchart.

**3. Results**

Following the first iteration, winglet taper ratio was found to have negligible effects on both aerodynamic efficiency and wing weight, and it was subsequently removed; thus, the design points were reduced to 648. Figure 8 is the plot of the calculated values of total drag ratio $C_D/C_{Dref}$ and wing weight ratio $W/W_{ref}$ of all the design points. The aircraft reference values are $C_{Dref} = 0.0641$ and $W_{ref} = 449.55$ kg.

The non-convergence area highlighted in Figure 8 was isolated and removed in subsequent analysis as convergence errors were identified during the aerodynamic calculations. Drag savings range from 0.05% to 5.105% at a weight penalty of 3% to a significant 35%. Comparable drag savings are obtained from a wide range of structural weights, which show the importance of an optimized structure. The study focused on cant and span, as they are the most significant drivers in terms of aerodynamic efficiency and weight.

A positive trend between the non-dimensional WRBM and the wing weight has been observed (Figure 9).

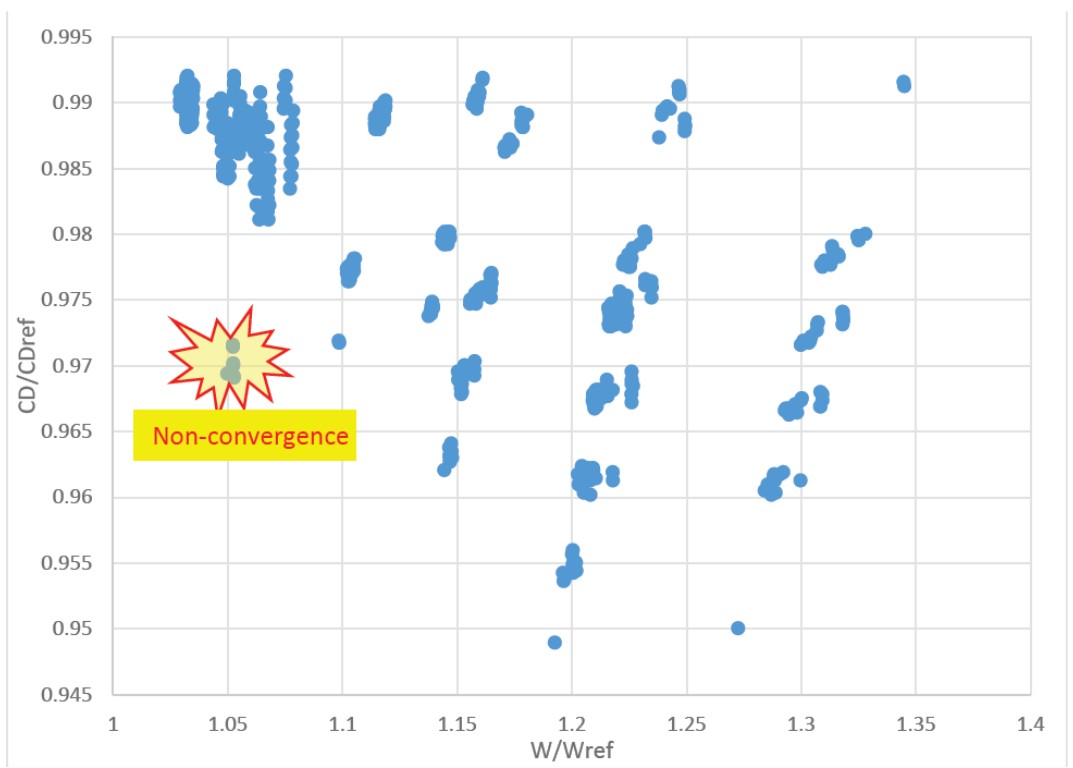

**Figure 8.** Design point values for drag and wing weight ratio.

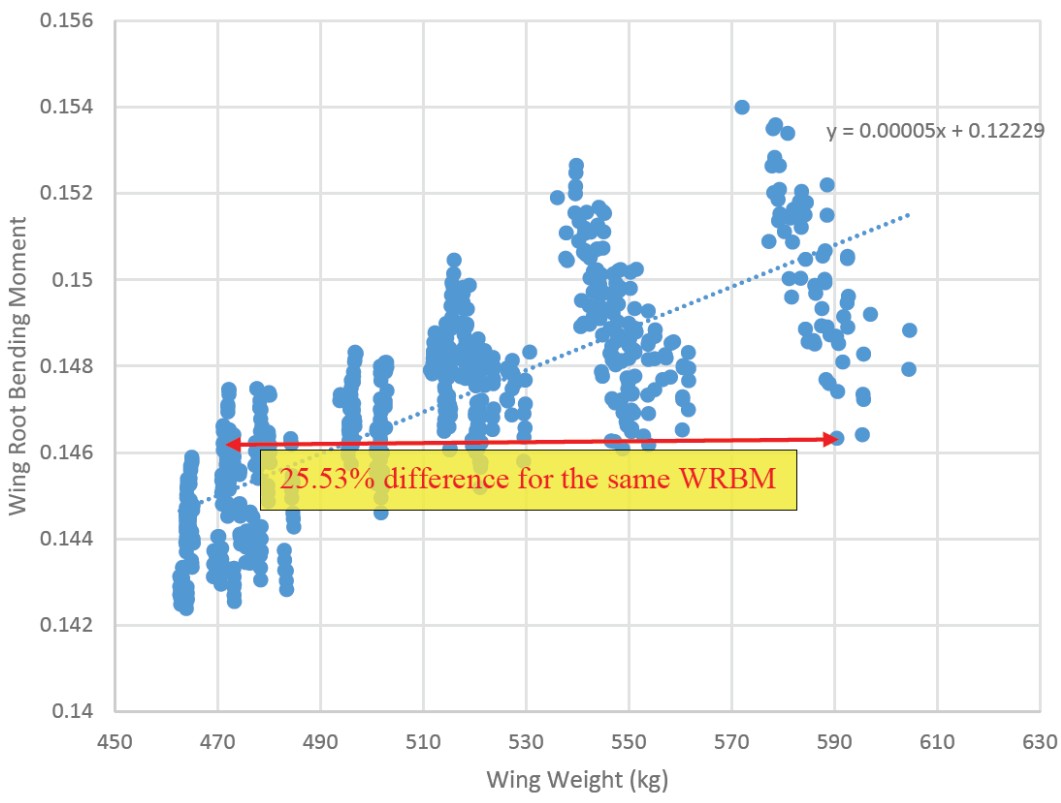

**Figure 9.** Wing-root bending moment (WRBM) vs. wing weight.

### 3.1. Cant Angle Effects

Designs with 80 degrees of cant experienced the lowest drag reduction for any combination of the other five parameters in comparison to a planar extension (7 degrees) or 43.5 degrees of cant angle (Figure 10). The spread in drag and wing weight increases with decreasing cant angles, and for a cant angle of 43.5 degrees, the maximum drag saving of 5.11% is achieved at a penalty of a 19.26% increase in wing weight. For 7 degrees, the respective numbers are 5% for the maximum drag saving at a 27.26% weight penalty. The cant effect on drag is positioned within the range of results from previous studies that compared planar to non-planar tip devices [9–15].

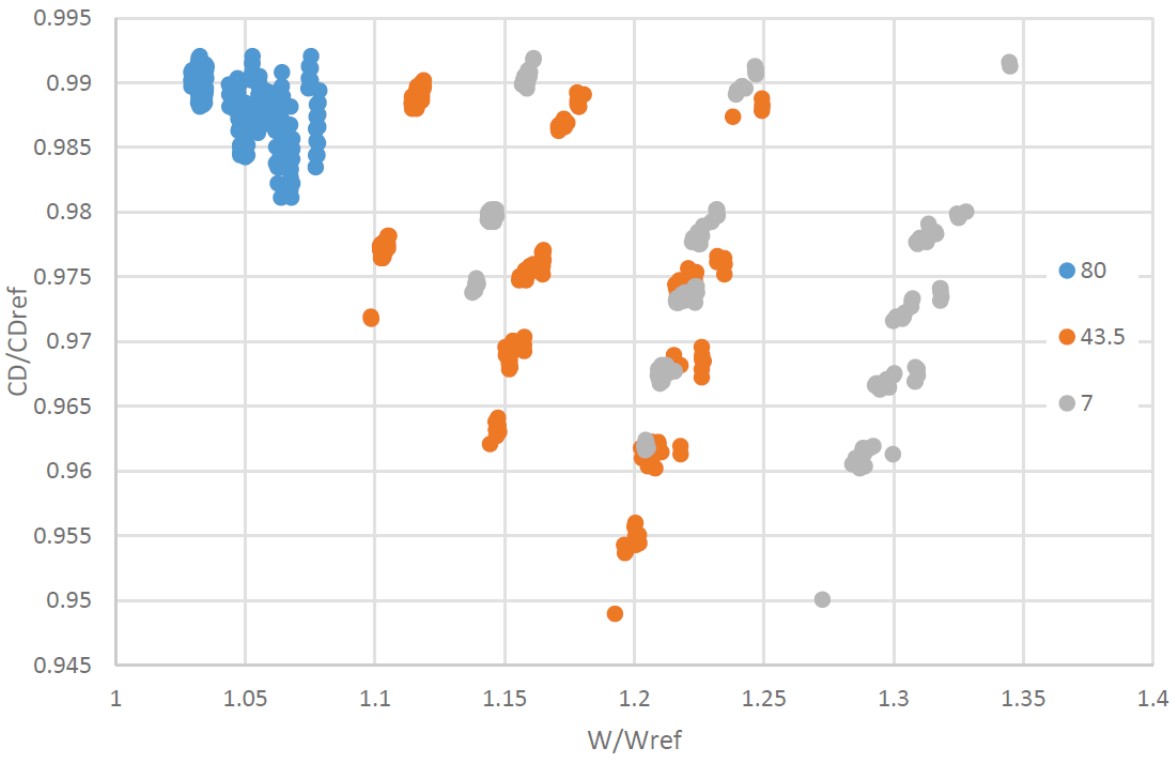

**Figure 10.** Cant angle distribution at the design space.

Although the designs were constrained by $C_{Lcruise}$, resulting in equal aerodynamic force and moment distribution, at lower cant angles, the bending load increases due to the increased bending arm, spanwise distance from the root. Furthermore, highly canted surfaces do not produce as much 'lift' in its traditional sense measured as a force in the Z-axis; instead, they will contribute more to a 'sideforce', even though the net force normal to the wing surface is constant. Since EMWET only uses lift force in the Z-axis as an input, the net bending effects might not have been fully accounted for the 80-degree designs during the structural sizing, as most of the force is perpendicular to the lift (normal to the wing surface). Therefore, it provides an explanation for the insensitivity to the weight change. Another simplification assumed with EMWET was modeling the wing and the winglet as a single unity, resulting in the tool-adding material across the span of the wing. The tool is programmed to add material inboard whenever possible to minimize outboard stress concentrations and take advantage of the root section's higher second moment of area to resist bending loads.

### 3.2. Span Effects

Figure 11 is the same as Figure 10 with shaded iso-span regions of 0.1, 0.15, and 0.2. For each cant angle, the drag savings are higher for increasing span ratios, incurring a higher weight penalty. For 80-degree cant, the distribution of points appears to be near vertical with each span ratio stacking in columns against each other. For each span "column", the range between the highest and lowest drag

range increase as the span increases. At a 0.1 span ratio, the 7-degree and 43.5-degree columns result in the same behavior, albeit with an offset in the X-axis with the 7-degree "column" resulting to be heavier and an offset in the Y-axis with decreased drag reduction effects. The same behavior can be observed for the 0.15 and 0.2 span ratios.

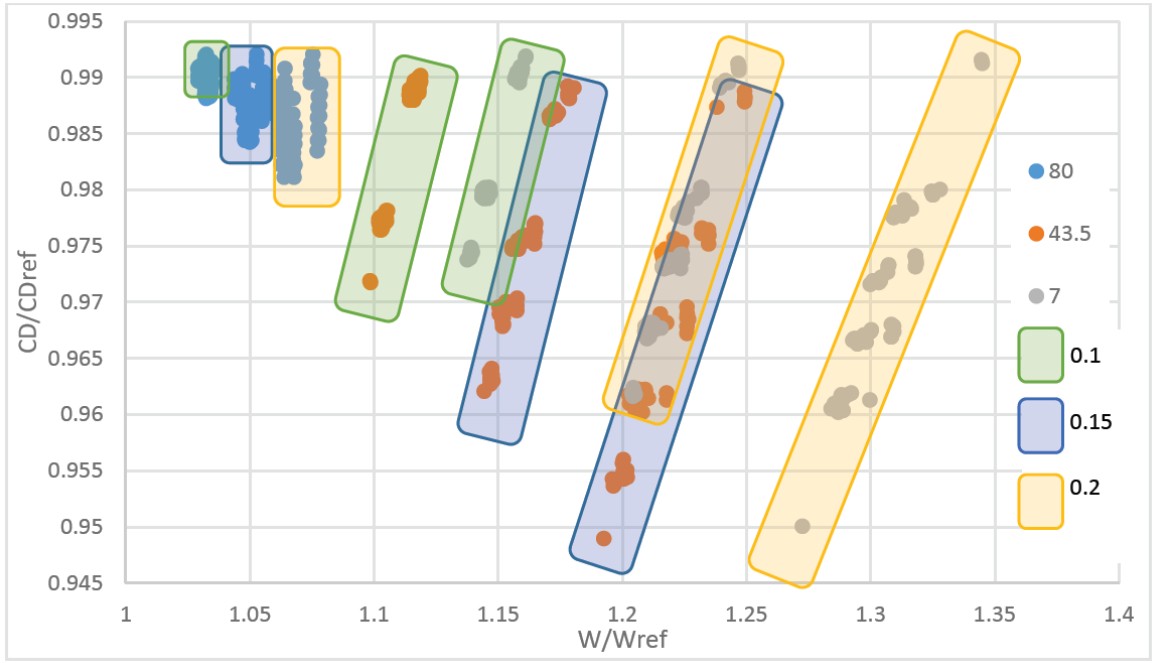

**Figure 11.** Winglet span effect on design point distribution.

### 3.3. Pareto Front

Plotting the result of two conflicting objectives, the drag and weight on each axis resulted in the formation of a Pareto front delimiting the feasible and unfeasible region of the design space (Figure 12). The Pareto front is formed from all Pareto efficient design points where the design points are found to be optimal to both objective functions without being able to improve one criterion without sacrificing the second criterion. The Pareto front consists of 80-degree cant angle designs with a maximum drag reduction of 1.89%, and the rest formed by 43.5-degree cant angle designs with a maximum drag reduction of 5.11%. As observed in Figure 12, the nature of the problem makes the Pareto front non-convex, and this creates the challenge that not all Pareto-optimal solutions can be obtained by using weighted sum approaches, which is the simplest and most straightforward way of obtaining multiple points on the Pareto-optimal front [43]. For the purpose of the performance assessment of our model, the Pareto optimal solution that provided the maximum cruise range was selected.

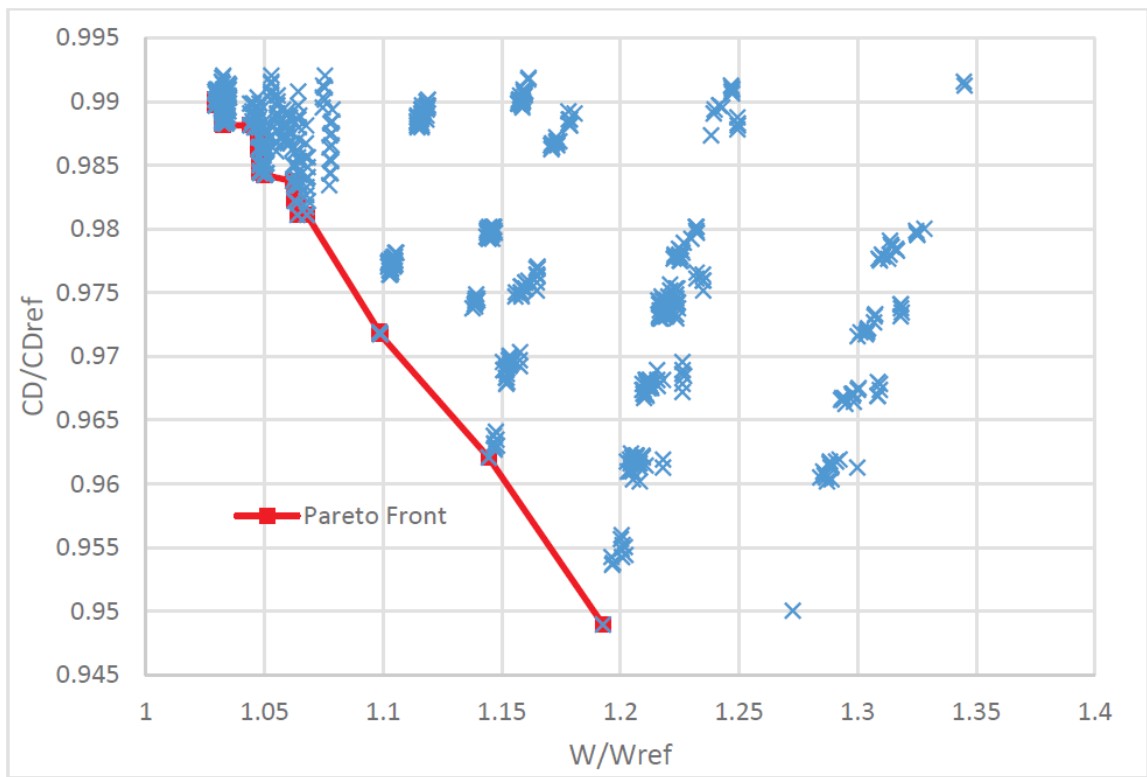

**Figure 12.** Pareto front in the design space.

### 3.4. Optimized Winglet Design

When assessing the best cruise range for the design points, a Pareto optimal design point for maximum cruise range was identified with a drag reduction of 1.19% and weight increase of 3.25%. The optimal winglet parameters were found to be the following (Table 11).

**Table 11.** Optimal winglet parameters.

| Parameters | Value |
|---|---|
| Winglet position | 0.8 |
| Cant | 80 (deg.) |
| Span | 0.1 |
| Sweep | 26 (deg.) |
| Toe-out | −1 (deg.) |
| Twist | −1 (deg.) |
| CD/CDref | −1.186% |
| W/Wref | 3.25% |

An isometric view of the optimal winglet is shown in Figure 13.

The optimal winglet configuration achieved a cruise range increase of 2.38%. The overall effect of the optimal configuration to the payload–cruise range diagram is shown at the Figure 14.

Regarding the aerodynamic characteristics of the optimal winglet configuration, the lift coefficients are nearly identical to the reference configuration. The lift-to-drag ratio exhibits an average increase of 7.2% across the polar due to the reduction of the induced drag, and subsequently, the overall drag. The structural impact of the optimal winglet configuration was isolated to the upper and lower skins without any changes to the spars. The reinforcements required on the upper skin cover most of the span of the wing, while at the lower skin, they cover just over half of the wing, all in all resulting in a total weight increase of 14.61 kg/wing (3.25% increase).

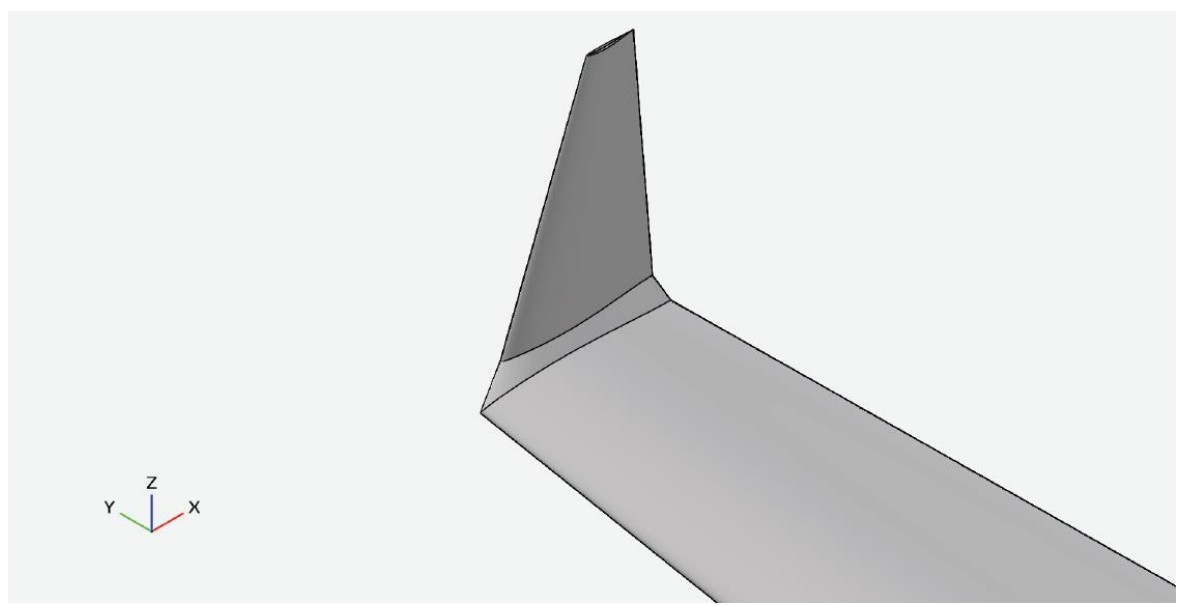

**Figure 13.** Isometric view of the optimal winglet.

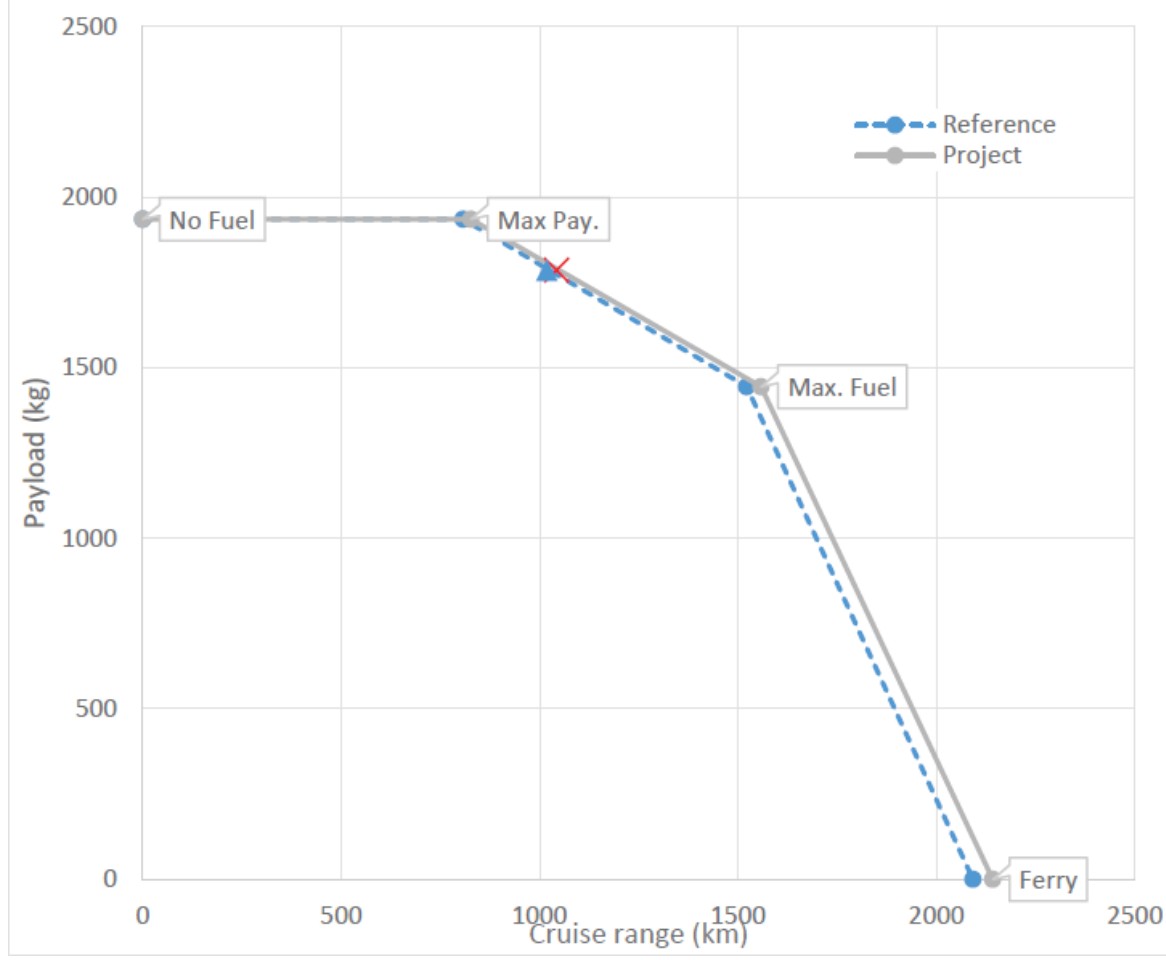

**Figure 14.** Cruise–range diagram for the reference wing and the optimal winglet (project) configuration.

## 4. Discussion

Similar to the considerations for which the C-130 aircraft type was not recommended as a potential platform for winglet retrofit [17], the Jetstream 31 has a relatively high aspect ratio of 10; with non-swept wings, it flies at relatively low cruise altitudes, and has the typical range of a regional turboprop aircraft. The combination of those design and operational parameters does have an effect on the expected winglet benefits. Work by Gautham and Bibin [44] suggests that the use of winglets improves the aerodynamic efficiency at all aspect ratios. However, the key finding of their work is that there exists an optimal aspect ratio at which winglets offer maximum effectiveness for a given flight condition. From an operational point of view, their work encourages retrofitting the aircraft with winglets, but at the same time, it underlines the importance of incorporating the winglet design and optimization as an integral part of the early conceptual design of a new aircraft platform.

The absence of commercial operational data for aircraft of similar size and engine types as the Jetstream 31 sets some validation challenges to the current study. One study of a similar, though conceptual, aircraft type [20] shows no considerable benefits for the cruise condition when using an optimized winglet that has a height of 10% of the wing semispan and a cant angle of 80 degrees. Instead, [20] reports significant benefits for the take-off, approach, and landing segments of the flight, since induced drag is the dominant drag component at those stages, and since turboprop aircraft will spend typically a higher percentage of their mission time climbing and approaching than other passenger aircraft, which operate at higher subsonic speeds and are powered by turbofan engines.

For a typical passenger aircraft configuration flying at higher subsonic speeds than the Jetstream 31 of the present study, significant aerodynamic and structural benefits have been observed for the cruise condition by using the 'curved winglet concept' suggested by Gueraiche and Popov [45]. Their proposed winglet design is considered a fair compromise between classic, low cant angle Whitcomb winglets and 'lifting' large cant angle winglets. Eliminating the constraints set by the cant angle paves the way to the exploration of 'variable cant angle' winglet concepts which, according to Guerrero et al. [46], can potentially enable aircraft designs to achieve optimal performance at a wide range of angle of attack values. Promising results have been also demonstrated by the 'Shark' wing-tip family [47].

## 5. Conclusions

Having reviewed the literature, no definite consensus exists on the benefits of non-planar over planar extensions, and the current work concludes that the use of a winglet with moderate cant (43.5 degrees) has achieved the highest value of total drag savings: a 9% decrease when compared to the reference aircraft configuration. The developed model has predicted aerodynamic loads and coefficients, which match very well with published flight test data for the reference Jetstream 31 aircraft, while the estimated wing weight was accurate to 1% of the statistical wing weight fraction for small turboprop aircraft. The optimal winglet design for maximum cruise range performance was not found to be the one that provides the greatest drag reduction, but a design with 80 degrees of cant, which resulted in a 1.19% drag reduction at a penalty of a 3.25% wing weight increase. The optimal design has come out of a non-convex design space, and as such, a potential future work item for the present study would be to find a convex formulation of the problem that would be more stable and easy to solve.

Previous studies indicate that the winglets are most beneficial when operating at high-altitude long-cruise segments for transonic jets, the design characteristics of which, and especially the aspect ratio, do not resemble those of the Jetstream 31. With reported block fuel improvements of more than 3% for ranges of more than 700 nm when installing winglets to the Boeing 737-800 [22], the installation of the optimal winglet design of the current study to the Jetstream 31 might not be considered as cost-efficient. However, the decision of retrofitting should not take into consideration only the aerodynamic and structural aspects, and it should typically also consider a DOC assessment, for which the method followed at [48] is suggested. A study [49] that has identified a relationship amongst the Cost Per Flying Hour (CPFH) and aircraft design parameters has shown that the two design variables

that contribute the most to the CPFH are the maximum specific fuel consumption and the aircraft empty weight; thus, it is almost certain that a winglet retrofit will influence the operating cost of the aircraft.

**Author Contributions:** The authors contributed equally to the preparation of the article.

**Funding:** This research received no external funding.

**Acknowledgments:** The authors would like to thank Luuk van der Schaft, Odeh Dababneh, and Vishagen Ramasamy for contributing with ideas and advice. Special thanks as well to Ali Elham, who has provided the student version of 'EMWET'.

**Conflicts of Interest:** The authors declare no conflicts of interest.

## Acronyms

| | |
|---|---|
| AR | Aspect Ratio |
| BAe | British Aerospace |
| CFD | Computational Fluid Dynamics |
| CPFH | Cost Per Flying Hour |
| DOC | Direct Operating Cost |
| EMWET | Elham Modified Wing Weight Estimation Technique |
| ISA | International Standard Atmosphere |
| MAC | Mean Aerodynamic Chord |
| MTOW | Maximum Take-Off Weight |
| MZFW | Maximum Zero-Fuel Weight |
| NACA | National Advisory Committee for Aeronautics |
| NASA | National Aeronautics and Space Administration |
| NFLC | National Flying Laboratory Centre |
| OEW | Operating Empty Weight |
| RPM | Revolutions Per Minute |
| USAF | United States Air Force |
| VLM | Vortex Lattice Method |
| WRBM | Wing Root Bending Moment |

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
