# Peer review of "Conceptual Design and Performance Optimization of a Tip Device for a Regional Turboprop Aircraft"

_aerospace, doi:10.3390/aerospace6100107_

Round 1

Reviewer 1 Report

The paper presented a conceptual study on the potential benefits of retrofitting a certain aircraft type with winglets. The paper has been written in a nice format, is easy to read and the concept is well explained. Although the results of the study are not ground breaking, still the methods and the approach used for deriving those is worthwhile exhibiting to the research community.

Author Response

Thank you very much, much appreciating your time and your very positive comments.

Many regards

Ilias

Reviewer 2 Report

The article is well prepared and written.

Minor recommendation:

Regarding aircraft operating in the higher subsonic and transonic speed range, cf. page 23 - discussion - some references should be added for completeness, see e.g.:

Heller, Kreuzer, Diermeier: Development and integration of a new high performance wingtip device for transonic aircraft, ICAS-2002-1.2.2, 2002.

Author Response

We would like to thank you for your valuable time and the positive comments.

The suggested reference has been added to the attached revised manuscript.

Many regards,

Ilias

Reviewer 3 Report

In this paper, the authors investigate different parameters of a designed a wing tip for improving the aerodynamics performance of the British Aerospace (BAe) Jetstream 31. The paper is well-written on a topical area but some further comments are given here:

Some figures are not shown clearly in black-and-white display, suggest the authors to use different markers not different colors. In fig.5, there is a line presents Z-height distribution, but there is no explain about what is Z-height distribute. There is no interpretation for some figures, such as Fig.6 and Fig.7. For the optimisation, the authors concluded that based on the aerodynamic characteristic, the cruise range can be increased, but I think that as parameters of the wing tip is changed, the extra weight will be added, so maybe the total weight will influence the fuel consumption?

Author Response

Dear reviewer, we would like to thank you for your time reviewing our work and for your constructive comments. Please find responses below. An amended manuscript which reflects all reviewers' comments is attached as well.

-Thanks for the comment regarding the black and white figure. We are going to work closely with the editing team to include the best possible quality for the final publishable version of the article.

 - Z-height distribution is now explained at the revised manuscript, together with revised interpretation of figures 6 and 7.

 - The wing weight increase (and subsequently the aircraft weight increase) and its influence to the fuel consumption has been taken into account for all the winglet design point results, as illustrated and explained at the figures 8, 9, 10, 11, 12 and at the Table 11. The objective of the article was to find an optimal winglet design, which eventually has achieved the maximum range increase by reducing the total drag by 1.19% at a mass penalty (increased wing mass) of 3.25%.